



# Influence of ENSO on entry stratospheric water vapor in coupled chemistry-ocean CCMI and CMIP6 models

Chaim I Garfinkel[1], Ohad Harari[1], Shlomi Ziskin Ziv[1,2,3], Jian Rao[1,4], Olaf Morgenstern[5], Guang Zeng[5], Simone Tilmes[6], Douglas Kinnison[6], Fiona M. O'Connor[7], Neal Butchart[7], Makoto Deushi[8], Patrick Jöckel[9], and Andrea Pozzer[10,11]

[1]The Fredy and Nadine Herrmann Institute of Earth Sciences, Hebrew University of Jerusalem, Jerusalem, Israel.
[2]Department of Physics, Ariel University, Ariel, Israel
[3]Eastern R&D center, Ariel, Israel
[4] Key Laboratory of Meteorological Disaster, Ministry of Education (KLME) / Joint International Research Laboratory of Climate and Environment Change (ILCEC) / Collaborative Innovation Center on Forecast and Evaluation of Meteorological Disasters (CIC-FEMD), Nanjing University of Information Science & Technology, Nanjing 210044, China.
[5] National Institute of Water and Atmospheric Research, Wellington, New Zealand.
[6] National Center for Atmospheric Research, Boulder, Colorado, USA
[7] Met Office Hadley Centre, Exeter, UK.
[8] Meteorological Research Institute, Tsukuba, Japan.
[9] Deutsches Zentrum für Luft- und Raumfahrt (DLR), Institut für Physik der Atmosphäre, Oberpfaffenhofen, Germany.
[10]Max Planck Institute for Chemistry, Mainz, Germany
[11]International Centre for Theoretical Physics, Trieste, Italy

*Correspondence to:* Chaim I. Garfinkel (chaim.garfinkel@mail.huji.ac.il)

**Abstract.** The connection between the dominant mode of interannual variability in the tropical troposphere, El Niño Southern Oscillation (ENSO), and entry of stratospheric water vapor, is analyzed in a set of the model simulations archived for the Chemistry-Climate Model Initiative (CCMI) project and for phase 6 of the Coupled Model Intercomparison Project. While the models agree on the temperature response to ENSO in the tropical troposphere and lower stratosphere, and all models also

agree on the zonal structure of the response in the tropical tropopause layer, the only aspect of the entry water vapor with consensus is that La Niña leads to moistening in winter relative to neutral ENSO. For El Niño and for other seasons there are significant differences among the models. For example, some models find that the enhanced water vapor for La Niña in the winter of the event reverses in spring and summer, other models find that this moistening persists, while some show a nonlinear response with both El Niño and La Niña leading to enhanced water vapor in both winter, spring, and summer. Focusing on

Central Pacific ENSO versus East Pacific ENSO, or temperatures in the mid-troposphere as compared to temperatures near the surface, does not narrow the inter-model discrepancies. Despite this diversity in response, the temperature response near the cold point can explain the response of water vapor when each model is considered separately. While the observational record is too short to fully constrain the response to ENSO, it is clear that most models suffer from biases in the magnitude of interannual variability of entry water vapor. This bias could be due to missing forcing processes that contribute to observed variability in

cold point temperatures.



## 1 Introduction

Water vapor (WV) is the gas with most important greenhouse effect in the atmosphere, and the feedback associated with stratospheric water vapor in response to increasing anthropogenic greenhouse gas emissions is around half of that for global mean surface albedo and cloud feedbacks (Forster and Shine, 1999; Solomon et al., 2010; Dessler et al., 2013; Banerjee et al., 2019;

Li and Newman, 2020). The amount of water vapor entering the stratosphere also regulates the severity of ozone depletion (Solomon et al., 1986) and is important for other aspects of stratospheric chemistry (Dvortsov and Solomon, 2001). Hence, it is important to understand how the comprehensive models that are used for e.g. future ozone projections capture the processes regulating entry of stratospheric water vapor.

Lower stratospheric water vapor concentrations are mainly determined by the tropical temperatures near the cold point,

where dehydration takes place as air parcels transit into the stratosphere (Mote et al., 1996; Zhou et al., 2004, 2001; Fueglistaler and Haynes, 2005b; Fueglistaler et al., 2009; Randel and Park, 2019). Several different processes have been shown to influence these cold point temperatures, and the goal of this work is to revisit the influence of one of these processes - El Niño Southern Oscillation (ENSO) - on entry water vapor in the lower stratosphere.

El Niño (EN), the ENSO phase with anomalously warm sea surface temperatures in the tropical East Pacific, leads to a

warmer tropical troposphere and cooler tropical lower stratosphere (Free and Seidel, 2009; Calvo et al., 2010; Simpson et al., 2011), with the zero-crossing in the vicinity of the cold-point (Hardiman et al., 2007). In addition, EN leads to a zonal dipole in temperature anomalies near the tropopause, and in particular to a Rossby wave response with anomalously warm temperatures over the Indo-Pacific warm pool and anomalously cold temperatures over the Central Pacific (Yulaeva and Wallace, 1994; Randel et al., 2000; Zhou et al., 2001; Scherllin-Pirscher et al., 2012; Domeisen et al., 2019). In the tropical tropopause layer

(TTL), water vapor increases in the region with warm anomalies and decreases in the region with cold anomalies by $\sim 25\%$ (Gettelman et al., 2001; Hatsushika and Yamazaki, 2003; Konopka et al., 2016).

The net effect of these zonally asymmetric and symmetric changes on water vapor above the tropical cold point is complex. The two largest EN events in the satellite era (in 1997/1998 and in 2015/2016) were followed by moistening of the tropical lower stratosphere (Fueglistaler and Haynes, 2005a; Avery et al., 2017), and the ERA-5 reanalysis, which tracks satellite water

vapor well over the last few decades, also shows a clear moistening after the 1982/1983 event (figure 3 of Wang et al., 2020). Strong La Nina (LN) events in 1998/1999 and 1999/2000 also clearly preceded elevated water vapor concentrations in the tropical lower stratosphere. The net effect of more moderate events (either LN or EN) is unclear (Gettelman et al., 2001), though there are two studies indicating a possible nonlinear effect. First, Brinkop et al. (2016) found that a strong LN following a strong EN event leads to a prolonged period of enhanced water vapor, which is then followed by reduced water vapor after

ENSO returns to more neutral conditions. Second, Garfinkel et al. (2018) found that both strong EN and LN events lead to elevated water vapor concentrations as compared to neutral ENSO. In addition, there is a strong seasonal dependence of the effect of EN on stratospheric water vapor, with the increase in water vapor for EN and decrease for LN occurring mainly in boreal spring (Calvo et al., 2010; Garfinkel et al., 2013; Konopka et al., 2016).



The limited length of the data record, and the importance of other atmospheric processes (e.g. the Quasi-Biennial Oscillation) which may interact nonlinearly with ENSO (Yuan et al., 2014), limits the confidence with which observed variability during and following ENSO events can be unambiguously associated with ENSO. Several studies have used simulations from single models to try to understand the role of ENSO for entry stratospheric water vapor (Scaife et al., 2003; Garfinkel et al., 2013; Brinkop et al., 2016; Garfinkel et al., 2018; Ding and Fu, 2018), though it is not clear whether the results are general to other models. The goal of this study is to consider a wider range of models, with a combined record length of over 2700 years, in order to better understand the response of stratospheric water vapor to ENSO. We focus here on chemistry-climate models, as these models must reasonably simulate entry water vapor otherwise their stratospheric chemistry will suffer from biases.

After introducing the data and methodology in Section 2, we contrast the impact of ENSO on stratospheric water vapor in 12 different chemistry climate models. Even though all models simulate a similar response to ENSO in the troposphere and also in the lower stratosphere (warming and cooling respectively), there is no consensus as to the impact of ENSO on stratospheric water vapor. Some models simulate enhanced water vapor for EN in both the winter of the event and the following spring, while other models find an opposite response, while some simulate a nonlinear response with both EN and LN leading to enhanced water vapor in spring (as is evident in GEOSCCM, Garfinkel et al., 2018). In all cases the temperature response near the cold point can explain the divergent responses of water vapor to ENSO.

## 2 Data and Methods

### 2.1 Data

We examine six models participating in the Chemistry-Climate Model Initiative (CCMI, Morgenstern et al., 2017) and six models participating in phase 6 of the Coupled Model Intercomparison Project (CMIP6 Eyring et al., 2016). However the focus in most of this paper is on the CCMI models for which data is archived at higher vertical resolution, as this allows for a more careful diagnosis of the physical processes. Coupled chemistry-climate models are expected to have more robust interannual variability of temperatures in the lower stratosphere as compared to models with fixed ozone (Yook et al., 2020), and hence we only include CMIP6 models with interactive stratospheric chemistry.

CCMI was jointly launched by the Stratosphere-troposphere Processes And their Role in Climate (SPARC) and the International Global Atmospheric Chemistry (IGAC) to better understand chemistry-climate interactions in the recent past and future climate (Eyring et al., 2013; Morgenstern et al., 2017). This modeling effort is an extension of CCMVal2 (SPARC-CCMVal, 2010), but utilizes up-to-date chemistry climate models that also include tropospheric chemistry. We consider the Ref-C2 simulations, which span the period 1960–2100, impose ozone depleting substances reported by the World Meteorological Organization (2011), and impose greenhouse gases other than ozone depleting substances as in Representative Concentration Pathway (RCP) 6.0 (Meinshausen et al., 2011). The full details of these simulations are described by Eyring et al. (2013). Note that the GEOSCCM simulations provided to CCMI did not have a coupled ocean, but Garfinkel et al. (2018) has already examined the ENSO-water vapor connection in this model in a coupled ocean configuration. As we are interested in connections between ENSO and the stratosphere, we only consider CCMI models with a coupled ocean in which ENSO develops spontaneously.



Table 1: Data products used

| data source | ensemble members | reference |
|---|---|---|
| ERA-5 | 1 | Hersbach et al. (2020) |
| NIWA-UKCA | 5 | Morgenstern et al. (2009) |
| CESM1 WACCM | 3 | Garcia et al. (2017) |
| CESM1 CAM4-chem | 3 | Tilmes et al. (2016) |
| HadGEM3-ES | 1 | Hardiman et al. (2017) |
| MRI-ESM1r1 | 1 | Yukimoto et al. (2012) |
| EMAC-L47MA | 1 | Jöckel et al. (2016) |

**Table 1.** The data sources used in this study.

The CCMI models used in this study are listed in Table 1. Harari et al. (2019) showed that each of these models simulate surface temperature variability in the Nino3.4 region similar to that observed.

In addition to the CCMI models, we also consider six Earth System models with coupled chemistry that are participating in CMIP6: CESM2-WACCM (Gettelman et al., 2019), GFDL-ESM4 (Dunne et al., 2019), CNRM-ESM2-1 (Séférian et al.,
2019), GISS-E2-1-G (Kelley et al., 2019), MRI-ESM2-0 (Yukimoto et al., 2019), and UKESM1-0-LL (Sellar et al., 2019). For these models we focus on the historical integrations of the period 1850 to 2014. Note that standard CMIP6 output includes the 70hPa and 100hPa levels but no level in-between, which limits our ability to diagnose physical processes near the cold point. All six of these models represent the Quasi-Biennial Oscillation (QBO) (Rao et al., 2020; Richter et al., 2020).

Model output is compared to water vapor in the ERA-5 reanalysis (Hersbach et al., 2020). Wang et al. (2020) has recently
shown that the ERA-5 reanalysis follows satellite water vapor closely over the last few decades, and we take advantage of the longer data record to provide a stronger constraint on the role of ENSO for water vapor.

## 2.2  Methods

This study focuses on the impact of ENSO on the stratosphere on interannual timescales, and in order to remove any impacts on longer timescales due to climate change, and also to remove any linear impacts from the Quasi-Biennial Oscillation
which is known to affect water vapor (Reid and Gage, 1985; Zhou et al., 2001, 2004; Fujiwara et al., 2010; Liang et al., 2011; Kawatani et al., 2014), we first use multiple linear regression to remove the linear variability associated with greenhouse gases and the QBO from all time series (i.e., the same regression is applied to temperature and water vapor). We use historical $CO_2$ concentrations for historical simulations and the equivalent $CO_2$ from the RCP6.0 scenario to track future greenhouse gas





concentrations (Meinshausen et al., 2011), and zonal averaged zonal winds from 5°S to 5°N at 50hPa with a 2 month lag to track the QBO. For consistency, this same multiple linear regression procedure is applied to CCMI, CMIP6, and ERA-5 data.

Each CCMI model makes data available at different pressure or sigma levels, which limits the precision with which we can compare models. However differences in the pressure levels at which data are available are generally less than 2hPa, and we

consider anomalies of each model from its own climatology. When considering entry water vapor for CCMI we examine the level closest to 80hPa and when considering the cold point temperature we examine the level closest to 90hPa archived by each CCMI model. The specific levels chosen for each CCMI model are indicated on the figures.

For ENSO, we use surface air temperature in the region bounded by 5°S-5°N and 190°E-240°E (i.e., the Nino3.4 region), as sea surface temperature was not available for all models at the time we downloaded the data. A composite of EN events

is formed if the average temperature in the Nino3.4 region in November through February (NDJF) relative to each model's climatology exceeds 1K, while a composite of LN events is formed if the average temperature anomaly is less than -1K. All other years are categorized as neutral ENSO.

Statistical significance of the composite mean response to a given ENSO phase is determined using a Student-t test. The adjusted $R^2$ (eq 3.30 of Chatterjee and Hadi, 2012) is used to quantify the added value in using a polynomial best fit (e.g. $H_2O$

$\sim a*EN^2 + b*EN$) instead of a linear best-fit (e.g. $H_2O \sim c*EN$). The adjusted $R^2$ takes into account the likelihood that a polynomial predictor will reduce the residuals by unphysically over-fitting the data. The polynomial fit can be preferred if the adjusted-$R^2$ for the polynomial fit is larger by any amount as compared to the linear $R^2$, though we only show the polynomial fit if the adjusted R-squared exceeds the $R^2$ for a linear fit by 33%. Note that the 33% criterion is subjectively chosen, though results are similar for a slightly modified criterion.

## 3  Results

We begin with the water vapor response to ENSO in the WACCM simulation included in CCMI in Figure 1. At 90hPa and also at higher pressure levels (i.e., lower in the TTL), EN leads to enhanced water vapor and LN to reduced water vapor in both winter and spring. Convection can rapidly mix moist boundary layer air with the TTL (e.g. Levine et al., 2007). Above the cold point, however, the water vapor response is not significant in November and December, but then shows a distinct nonlinearity

in subsequent months, with both EN and LN leading to enhanced water vapor. This nonlinear effect is similar to that seen in the GEOSCCM model by Garfinkel et al. (2018).

These results are summarized in Figure 2a, which shows the water vapor response for EN (the events in the right shaded box on Figure 1), LN (the events in the left shaded box on Figure 1), and neutral ENSO (all other events). In January through June, both EN and LN lead to significantly more entry water vapor than neutral ENSO. The pronounced moistening during EN

peaks in the spring after the event has already begun to decay. These effects are all consistent with that seen in GEOSCCM in Garfinkel et al. (2018). A generally similar effect is evident in CAM4Chem, which shares code with WACCM.

The four models shown in Figure 2cdef have a qualitatively different response to ENSO than the NCAR models and GEOSCCM. Specifically, HadGEM3-ES, NIWA, MRI-ESM1r1, and EMAC-L47MA all simulate somewhat more water vapor





for LN than neutral ENSO (though this effect is generally not statistically significant), and significantly more water vapor for neutral ENSO than EN, in January through April. In NIWA and EMAC-L47MA this effect extends through all calendar months.

This large diversity in the entry water vapor response to ENSO occurs despite the fact that all models simulate a qualitatively similar response in tropospheric and lower stratospheric temperatures. Figure 3 shows the distribution of 15°S-15°N temperature as a function of longitude and height for these six models in March and April, the months with the strongest disparity among the models in the response of entry water to ENSO. All models are characterized by a more pronounced warming between 200°E and 250°E immediately above the region with warming sea surface temperatures as compared to other longitudes, and in all models there is a zonal mean increase in temperature throughout the troposphere. The tropospheric warming peaks in the upper troposphere, and extends up to the TTL near 120°E in all models. Furthermore, all models simulate a lower stratospheric cooling (above 70hPa) in response to EN and a warming in response to LN. While the magnitude of these features differs among the model, the patterns are robust.

Near the tropopause, however, there is less agreement among the models in the large scale temperature response, and this difference can account for the large diversity in the water vapor responses to ENSO. The middle column of Figure 2 shows the zonally averaged temperature response to ENSO in the tropics near 90hPa. The zonally averaged temperature response to ENSO in WACCM has little resemblance to the water vapor response. Rather, the water vapor response can be better understood by focusing on the coldest region of the tropics. Due to the relative slowness of vertical transport as compared to horizontal transport in the tropical tropopause layer, entry water vapor is sensitive to the coldest regions in the tropics and not just zonal mean temperatures (i.e. the cold point, Mote et al., 1996; Hatsushika and Yamazaki, 2003; Fueglistaler et al., 2004; Fueglistaler and Haynes, 2005a; Oman et al., 2008; Randel and Park, 2019). We quantify this effect as follows: We first sort the temperature in all grid points from 15°S to 15°N in each bimonthly period. We then calculate the threshold temperature associated with the first quintile, second quintile, etc., of tropical temperatures. We compute these quintiles separately for the EN, LN, and neutral ENSO, and then compute the difference for each ENSO phase from the model climatology. The results of this analysis for the second quintile are shown in the right column of Figure 2a. The coldest 20% of the tropics is ∼0.25K warmer during EN as compared to the model climatology from November through June, while for LN and neutral ENSO the coldest 20% of the tropics is colder than the model climatology. Overall, the correlation between the 20% quintile cold point temperature anomalies and the water vapor anomalies is 0.73 (Table 2). Results are generally similar for CAM4Chem through June: the correlation of entry water with the coldest 20% is positive, while the correlation with zonal mean temperatures is not.

HadGEM3-ES, NIWA, MRI-ESM1r1, and EMAC-L47MA all simulate similar temperature responses if we focus on the zonal mean or the coldest 20% of the tropics, though correlations with entry water vapor are higher if we focus on the coldest 20% of the tropics rather than zonal mean temperature (Table 2). For these models, temperatures are warmer for LN than neutral ENSO and colder for EN than neutral ENSO (Table 2). Overall, the temperature response to ENSO in the coldest 20% of the tropics near $90hPa$ can help account for the substantial inter-model diversity in the response of entry water to the stratosphere.



Garfinkel et al. (2013) and Ding and Fu (2018) considered the possibility that sea surface temperatures (SSTs) in the central Pacific may have a different effect on entry water than SSTs in the East Pacific, and the two studies, using different individual models, found that warmer SSTs in the central Pacific lead to dehydration. We evaluate this effect for the CCMI models in Figure 4. Specifically, the left column of Figure 4 shows the correlation between entry water in March and April and near

surface temperature in January and February. There is clearly a wide range of responses evident, and consistent with Figure 2, some models show a positive correlation between SSTs in the Nino3.4 region (e.g. WACCM) while others show a negative correlation (HadGEM3-ES, NIWA, MRI-ESM1r1, and EMAC-L47MA). There is no clear difference in the correlation between near surface temperature to the east or west of the Nino3.4 region (indicated with a black box on Figure 4), and clearly there is no consensus among the models as to whether warmer SSTs in the central Pacific lead to dehydration.

Dessler et al. (2013) and Dessler et al. (2014) find that tropical tropospheric temperatures at 500hPa is a better predictor of entry water vapor than ENSO in the satellite record. We therefore consider for each model the correlation between entry water in March and April and 500hPa temperature in January and February in Figure 4 (right column). There is clearly a wide range of responses evident, and the response is similar in pattern to that in 4a-f. Specifically, some models show a positive correlation of entry water with mid-tropospheric temperatures (e.g. WACCM and CAM4Chem) while others show a negative

correlation (HadGEM3-ES, NIWA, MRI-ESM1r1, and EMAC-L47MA). Note that all models simulate a long-term moistening trend of the lower stratosphere if the trend is computed before applying the MLR described in section 2 (trend indicated above Figure 4ghijkl), and of the six models considered, the two with the strongest long-term moistening trend simulate a negative correlation between temperatures at 500hPa and entry water vapor when focusing on interannual variability. Hence there is no evidence that temperatures at 500hPa are a more discriminatory predictor of entry water vapor on interannual timescales than

ENSO. That being said, it is conceivable that on longer timescales, the magnitude of mid-tropospheric warming would be e.g. related to an upward expansion of the TTL (a robust response to climate change) and such an expansion of the TTL might be expected to lead to more entry water vapor. A thorough investigation of this possibility is beyond the scope of this paper.

## 4   Comparison to observations and CMIP6

What is the observed response of entry water vapor to ENSO? Figure 5a is as in Figure 2a but for ERA5 entry water vapor,

and while both LN and EN are associated with more water vapor, the difference between EN and neutral ENSO and between LN and neutral ENSO is not statistically significant except in July and August. Similarly, the regression coefficient of a linear best-fit of entry water vapor with ENSO (as in the black line on Figure 1a) is also not statistically significant except in July and August (details not shown). Despite the lack of a significant effect in observations, the models that appear to be closest to the observed response are the NCAR models and also the GEOSCCM simulations evaluated by Garfinkel et al. (2018).

A complication when comparing the models to ERA5 entry water is that ∼140 years at least of model data are available for each model while only 41 years of data, and nine EN and nine LN events, are available for observations. Hence it is ambiguous whether the difference between models and observations reflects an actual model bias, or alternately might reflect uncertainty given the small observational sample (i.e. the large error bars on Figure 5a overlap the error bars on Figure 2 for





many models). In order to better compare model and observations, we adopt an Monte Carlo subsampling technique. Taking EN as an example, we randomly select 9 EN events from each model to match the number of observed EN events in the ERA5 period, and compute the mean entry water vapor anomaly for these events. We then repeat this random sampling 2000 times, with different EN events randomly included in the subsample. Finally, we compute the top and bottom 2.5% quantiles of the

subsampled response to EN, to which we can compare the observed response.

Figure 5b-g show the response to ENSO in these subsamples for each model, and we repeat the observed response with a thin line. If the observed response falls outside of the middle 95% of the subsampled response (indicated with a vertical line), then the model response to ENSO is inconsistent with that observed. Of all models considered in this paper, HadGEM3-ES is most consistent with observations, as the subsamples of the model overlap the observed response in nearly all seasons and

phases (Figure 5d). The WACCM and CAM4Chem response to ENSO is consistent with observations in JF, MA, and MJ, though not in other seasons (Figure 5bc). The other models, however, suffer from large discrepancies between the observed and modeled responses to ENSO even when we compare similar sample sizes.

An additional metric to evaluate differences in observed vs. modeled ENSO teleconnections is for the model to simulate a similar amount of variance as compared to that observed, as otherwise the model does not satisfactorily capture internal

atmospheric variability (Deser et al., 2017; Garfinkel et al., 2019; Weinberger et al., 2019). We therefore compare the standard deviation of entry water vapor for each model in Figure 6a. The 95% confidence interval of the standard deviation as given by a chi-squared test is indicated with a vertical line. In boreal winter, only HadGEM3-ES and MRI-ESM1r1 simulate realistic variability, with NIWA simulating too much (likely because it has the warmest climatological cold point of any model) and the other models simulating too little. In boreal summer, all models suffer from unrealistic variability.

Recently, at least six coupled ocean-chemistry climate models have participated in CMIP6, and we now assess the ENSO-water vapor connection in these models: CESM2-WACCM, GFDL-ESM4, GISS-E2-1-G, MRI-ESM2-0, UKESM1-0-LL, and CNRM-ESM2-1. Of these six models, three are newer versions or successors of models that participated in CCMI (CESM2-WACCM, MRI-ESM2-0, and UKESM1-0-LL). Figure 7 is as in Figure 5 but for 70hPa water vapor, as water vapor near 80hPa is not a standard CMIP6 output variable. The observed water vapor response at 70hPa resembles that at 82hPa (Figure 7a

vs. Figure 5a). While the models generally agree that LN leads to moistening in winter, the models simulate a wide diversity of responses in the spring and summer following LN and EN. For only one model is the modelled response consistent with observations in that the subsampled response from the model encompasses observations (CESM2-WACCM). For all other models the observed and modeled response to water vapor are inconsistent in at least one season and one ENSO phase, and while the inconsistency is relatively small for GFDL-ESM4 and GISS-E2-1-G, it is pronounced for CNRM-ESM2-1, MRI-

ESM2-0, and UKESM1-0-LL.

The standard deviation of 70hPa tropical water vapor for each CMIP6 model is shown in Figure 6b. While nearly all CCMI models struggled to capture realistic variability, half of the CMIP6 models simulate a realistic amount of variability. Specifically, the CCMI models HadGEM3-ES and MRI-ESM1r1 failed to simulate realistic variability in spring, but the corresponding CMIP6 models UKESM1-0-LL and MRI-ESM2-0 are realistic. GISS-E2-1-G also simulates a realistic amount of variability.





However the other three CMIP6 models simulate too-little variability, though the bias in WACCM is smaller in the CMIP6 CESM2-WACCM than in the CCMI version of WACCM.

The lack of a realistic level of entry water vapor variability in the CCMI models may be due to (at least) two potentially complementary factors. First, these models may not adequately simulate all of the processes leading to observed variability (e.g. ice lofting). Second, the too-low variability in most models may be due to the models not including all of the relevant forcing processes (e.g. aerosols in the Asian monsoon) that contribute to observed variability. Future work to isolate the importance of each factor is clearly needed.

## 5 Summary

The amount of water vapor entering the stratosphere helps to determine the overall greenhouse effect and also regulates the severity of ozone depletion. The goal of this study is to understand how the comprehensive models that are used for e.g. future ozone projections capture the connection between the dominant mode of interannual variability in the tropical troposphere, El Niño Southern Oscillation (ENSO), and entry of stratospheric water vapor.

All models simulate a warmer tropical troposphere and cooler tropical lower stratosphere for El Niño (EN), the ENSO phase with anomalously warm sea surface temperatures in the tropical East Pacific (consistent with previous modeling and observational studies, Free and Seidel, 2009; Calvo et al., 2010; Simpson et al., 2011). Furthermore, EN leads to a zonal dipole in temperature anomalies near the tropopause in these models, with anomalously warm temperatures over the Indo-Pacific warm pool and anomalously cold temperatures over the Central Pacific (again consistent with the observed effect and previous modeling studies, Yulaeva and Wallace, 1994; Randel et al., 2000; Zhou et al., 2001; Scherllin-Pirscher et al., 2012; Domeisen et al., 2019). This is the first multi-model study to explore the subsequent effects on water vapor. While nearly all models simulate a moistening for LN in winter and early spring as compared to neutral ENSO, for other seasons and for EN we find complex changes that differ in sign among the models. Some models simulate enhanced water vapor for EN in both the winter of the event and the following spring, other models find an opposite response, while some show a nonlinear response with both EN and LN leading to enhanced water vapor in spring. A similarly wide diversity of responses is evident if we focus on Central Pacific ENSO versus East Pacific ENSO, or temperatures in the mid-troposphere as compared to temperatures near the surface. Despite this diversity in response, the temperature response near the cold point can explain the response of water vapor when each model is considered separately, with the response of temperatures in the coldest 20% of the tropics to ENSO able to explain the simulated response to water vapor.

The observational record is too short to confidently classify models as "good" or "bad", though some models simulate a response inconsistent with that observed even if we subsample their output to mimic the length of the observational record. Furthermore, nearly all CCMI models and half of the CMIP6 models suffer from biases in the amount of interannual variability in entry water vapor, with most models simulating too little variability. This bias could be due to biases in how the models simulate key processes regulating water vapor, though the close correspondence between temperatures in the coldest 20% of the tropics and the simulated water vapor response to ENSO (Table 2) suggests that the models resolve the most important





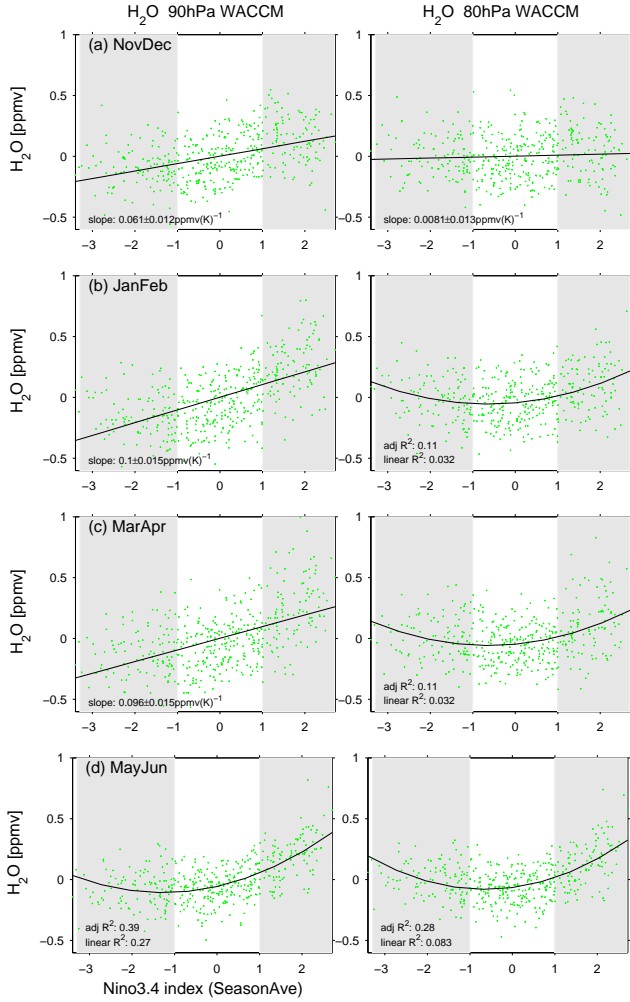

**Figure 1.** Anomalous 90hPa and 80hPa water vapor in WACCM as compared to the value of the Nino3.4 index for (a) November and December; (b) January and February; (c) March and April; (d) May and June. Each dot corresponds to one model-year. When a polynomial fit better describes the dependence on ENSO than a linear fit, we show the $R^2$ for a linear fit and adjusted $R^2$ for the polynomial fit (see section 2.2). Otherwise we show a linear least-squares best fit in each panel.

factor governing entry water vapor variability (Mote et al., 1996; Hatsushika and Yamazaki, 2003; Fueglistaler et al., 2004; Fueglistaler and Haynes, 2005a; Oman et al., 2008; Randel and Park, 2019). Alternately, the models could be missing forcing processes that contribute to variability in cold point temperatures even though historical emissions were input to the models. The good news is that all three modeling groups that contributed to both CCMI and CMIP6 show an improvement in this bias.

5  Future work is needed to fully consider what led to this improvement.



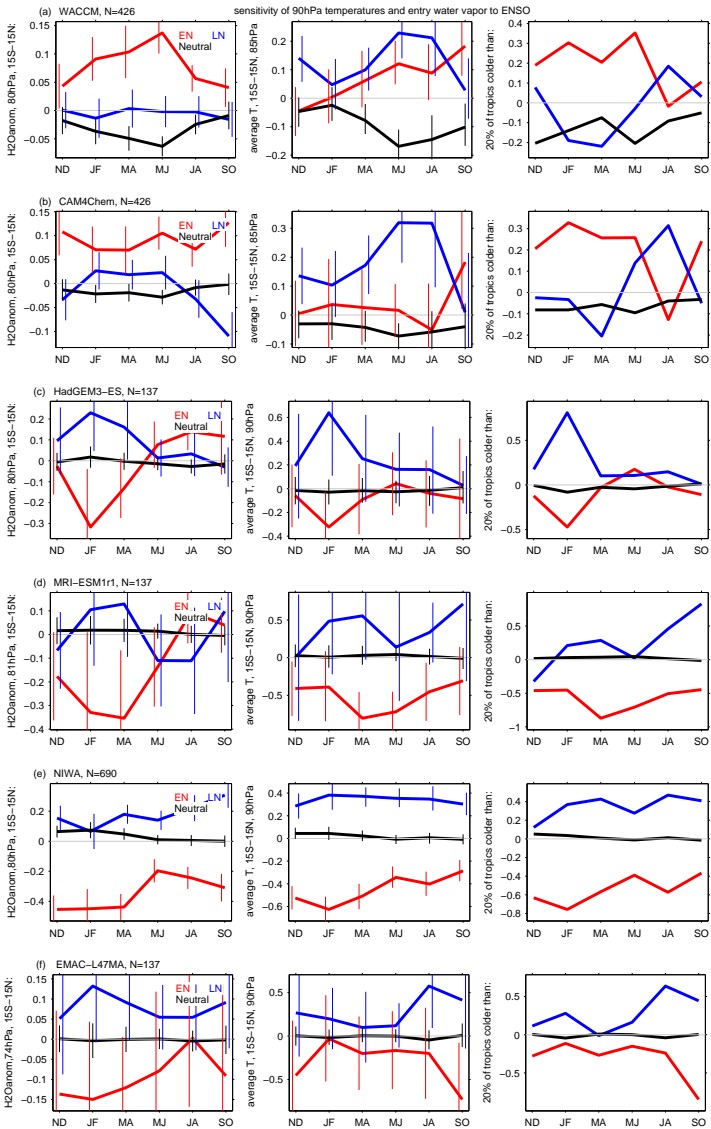

**Figure 2.** (left) Tropical water vapor from 15S-15N near 80hPa in each of the 6 CCMI models considered here from the late fall as the event is developing through the following summer for (red) El Nino, (blue) La Nina, (black) neutral ENSO. 5% confidence intervals on the anomalous response based on a two-tailed Student-t test are shown. (middle) response of zonally averaged temperature anomalies from 15S-15N near 90hPa for each model. (right) evolution of the temperature of the coldest 20% of the tropics at 90hPa for each model in each ENSO phase as compared to the model's climatology.



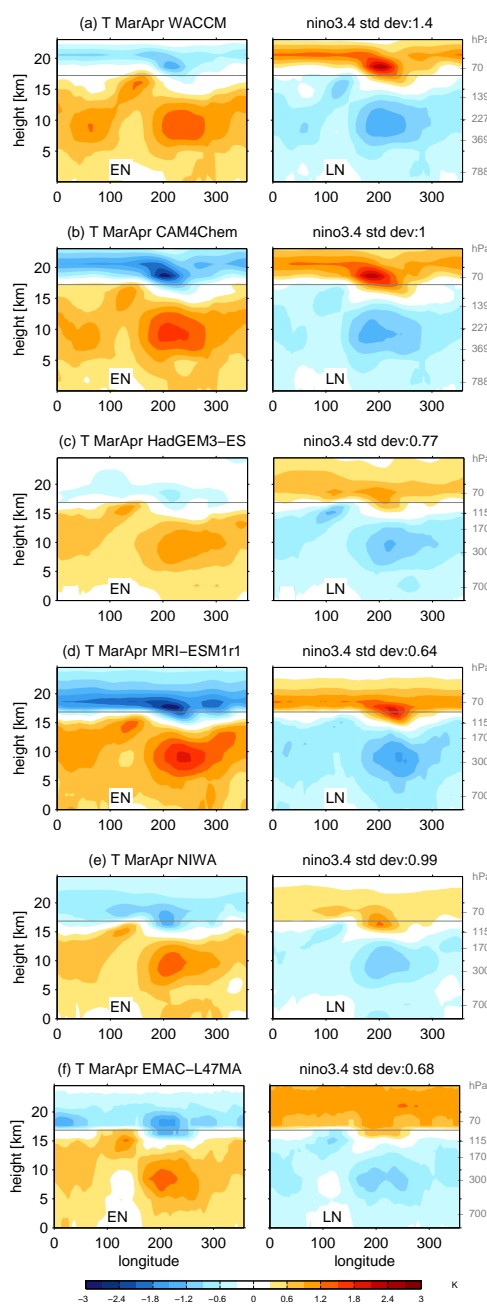

**Figure 3.** Longitude versus height cross section of the 15S-15N temperature anomalies during (left) El Nino, (right) La Nina in each of the CCMI models



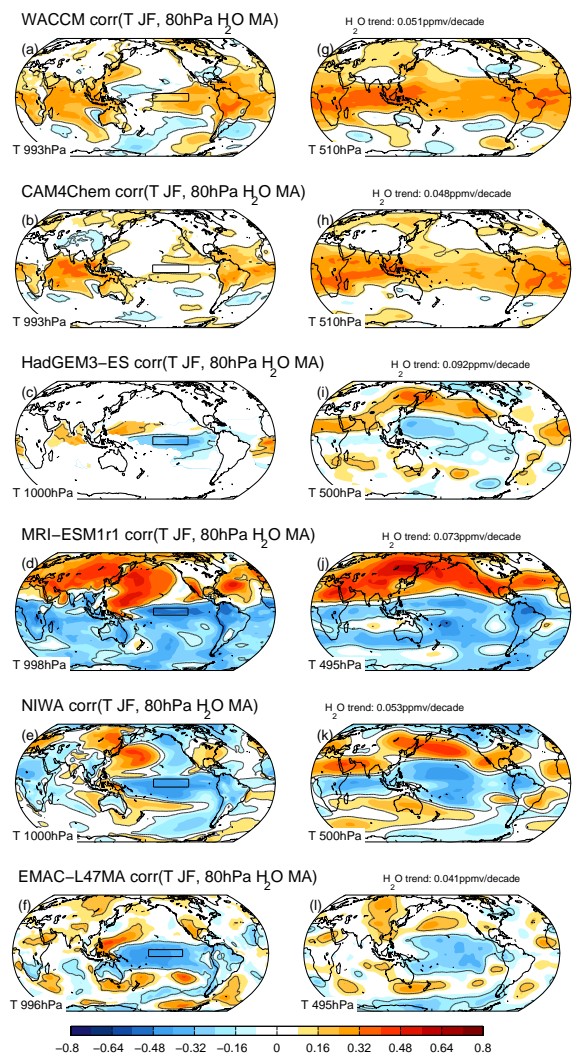

**Figure 4.** Correlation of (left) near-surface temperature and (right) temperature near 500hPa with entry water vapor in each of the CCMI models, with temperature taken for January and February and water vapor in March and April. A black line indicates correlations significantly different from zero at the 95% confidence level.





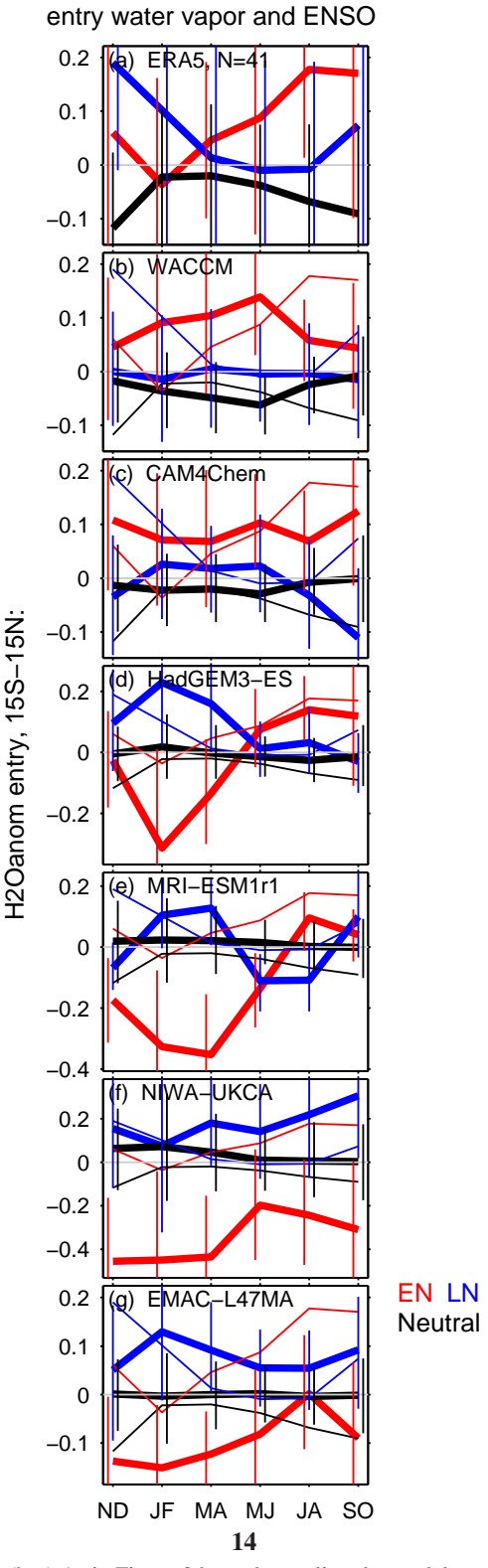

**Figure 5.** (a) As in Figure 2a but for ERA5, (b-g) As in Figure 2 but subsampling the model output for each model to match the sample size in observations for each ENSO phase. The uncertainty (marked with vertical lines) of the model response is determined by the Monte Carlo subsampling as described in the text.




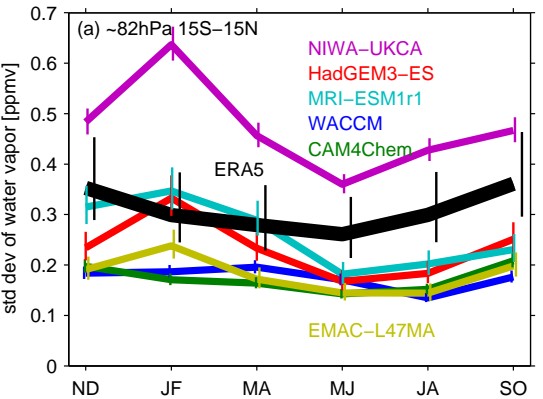
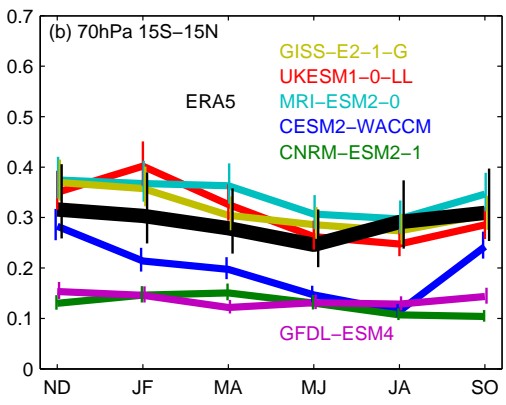

**Figure 6.** Standard deviation of tropical entry water vapor for each model in (a) CCMI near 82hPa and (b) CMIP6 at 70hPa, and for ERA5 water vapor (thick line). The vertical lines denote 95% confidence bounds as discussed in the text.





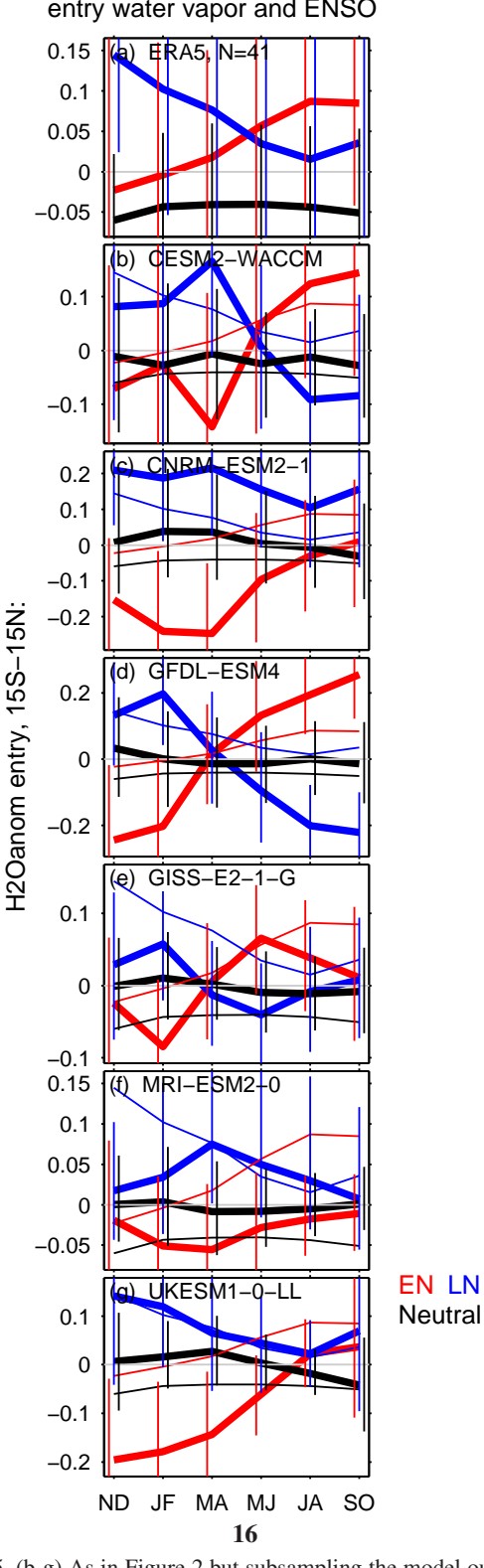

**Figure 7.** (a) As in Figure 2a but for ERA5, (b-g) As in Figure 2 but subsampling the model output for six CMIP6 models with interactive chemistry to match the sample size in observations for each ENSO phase for water vapor at 70hPa. The uncertainty (marked with vertical lines) of the model response is determined by the Monte Carlo subsampling as described in the text.





Relationship between entry water vapor and cold point temperature

|  | zonal averaged T, 15S-15N | 20% quintile |
|---|---|---|
| WACCM | -0.28 | 0.73 |
| CAM4Chem | -0.26 | 0.41 |
| HadGEM3-ES | 0.42 | 0.75 |
| MRI-ESM1r1 | 0.50 | 0.58 |
| NIWA-UKCA | 0.54 | 0.96 |
| EMAC-L47MA | 0.35 | 0.69 |

**Table 2.** Note that the pressure level for each model differs due to data availability, and the levels used for this chart are indicated on Figure 2.



## 6   acknowledgment

CIG was supported by an European Research Council starting grant under the European Union's Horizon 2020 research and innovation programme (grant agreement No 677756). We thank the international modelling groups for making their simulations available for this analysis, the joint WCRP SPARC/IGAC CCMI for organizing and coordinating the model data analysis
activity, and the British Atmospheric Data Centre (BADC) for collecting and archiving the CCMI model output. All datasets used in this study are available online: |http://blogs.reading.ac.uk/ccmi/badc-data-access|. Correspondence and requests for data should be addressed to C.I.G. (email: chaim.garfinkel@mail.huji.ac.il).

F.M. O'Connor and the development of HadGEM3-ES was supported by the joint DECC/Defra Met Office Hadley Centre Climate Programme (GA01101) and by European Commission's 7th Framework Programme, under grant agreement no.
603557 (StratoClim project).

The EMAC simulations have been performed at the German Climate Computing Centre (DKRZ) through support from the Bundesministerium für Bildung und Forschung (BMBF). DKRZ and its scientific steering committee are gratefully acknowledged for providing the HPC and data archiving resources for the consortial project ESCiMo (Earth System Chemistry integrated Modelling).

The CESM project is supported primarily by the National Science Foundation (NSF). This material is based upon work supported by the National Center for Atmospheric Research, which is a major facility sponsored by the NSF under Cooperative Agreement No. 1852977. Computing and data storage resources, including the Cheyenne supercomputer (doi:10.5065/D6RX99HX), were provided by the Computational and Information Systems Laboratory (CISL) at NCAR.

OM and GZ acknowledge the UK Met Office for use of the Unified Model, the NZ Government's Strategic Science In-
vestment Fund (SSIF), and the contribution of NeSI high- performance computing facilities to the results of this research (https://www.nesi.org.nz). OM also acknowledges funding by the New Zealand Royal Society Marsden Fund (grant 12-NIW-006). MD acknowledges funding by the Japan Society for the Promotion of Science (grant numbers: JP20K04070).



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
