# Peer review of "Influence of ENSO on entry stratospheric water vapor in coupled chemistry-ocean CCMI and CMIP6 models"

_Atmospheric Chemistry and Physics, 2020_

## Referee Comment (RC1) · Anonymous Referee #1 · 21 Oct 2020

**General:**
This a very important and well-written paper and should be published by ACP. However, it contains some substantial inconsistencies related to the formulation of the results derived from pure model studies (like CCMI) or from models validated to some extent with observations (like the ERA5 reanalysis) or from pure observations (like MLS, HALOE or other satellite data). Thus, there are few major points which have to be clarified.

**Major points**

[Figure]

- You use water vapor from the ERA5 reanalysis to validate the CCMI/CMIP6 models...you call it in the title of section 4: "Comparison to observation". In relation to the temperature fields, e.g. to the cold-point tropopause, ERA5 can be understood as an assimilated data set (like most other reanalyses), but this is certainly not the case in relation to the stratospheric water vapor. The stratospheric water vapor is a product derived from the ERA5 chemistry-transport module where, of course, the resolved cold-point tropopause plays a vary important important role. Thus, stratospheric water vapor in ERA5 is not assimilated with observations (like temperature), it is much more an almost "pure" model product.

  Typically, all older reanalyses (ERA-Interim, JRA-55) have stratospheric water vapor that is not good enough for any scientific interpretation. Davis at al., ACP, 2019 writes: "...because of the known deficiencies in the representation of stratospheric transport in reanalyses, the stratospheric water vapor products from the current generation of reanalyses should generally not be used in scientific studies." The improvement of ERA5 (as documented in Wang et al., 2020) is probably a consequence of a better transport scheme...You should mention all these points. Even if the stratospheric water vapor in ERA5 is to some extent validated (Wang et al., 2020), there is still not enough validation of ERA5 for the period 1979-1995, which was strongly influenced by volcanic eruption (El Chichon and Pinatubo) and for which satellite observations are either not available or strongly disrupt by volcanic aerosol. Finally, it is not clear if ERA5.0 or ERA5.1 is used (it is also not clear in Wang et al., 2020). The latter version removes a significant temperature bias of the cold point tropopause for the period 2000-2009 (see Simmons et al., 2020, ECMWF, Technical Memo 859).

- Your study shows that there are models which do not lead to a moistening of the stratosphere after El Nino events (like last 3 models shown in Fig. 5). However, moistening of the stratosphere after strong EL Nino is the best documented and validated finding (Geller et al., 2002, Scaife et al., 2003, Randel et al., 2009,

Konopka et al., 2016) although as pointed by Garfinkel et al. 2013, there are some differences in the intensity of such moistening which depend on maximum relative to the winter season and relative to to to El Nino's location (Central or Eastern Pacific). I think that this point (models do not represent moistening after EL Nino correctly) should be mentioned in the abstract. I think that this point is at least equally important as your statement related to the non-linear behavior between La Nina and El Nino or that in all models La Nina leads to moistening in winter relative to neutral ENSO. Both statements, even very interesting, are derived "only" from the models and only partially present in ERA5 reanalysis.

**Minor comments:**

- P2L3

  ...is around half of that for global mean surface albedo OR cloud feedbacks... (I think, this is what the the cited papers show)

- P2L28

  I think, it is not correct to use the Brinkop et al., 2016 citation to support the idea of nonlinear effects....what they showed is that the millennium drop was a combination of El Nino (i.e. wet phase in the stratosphere) followed by La Nina (i.e. dry phase in the stratosphere) with QBO being in the east phase (i.e. enhancing the dry phase)...This is a very linear interpretation without any non-linear effects

- P2L30

  For the non-linear effect discussed in Garfinkel et al., 2018 you should also mention that this is a pure model study without any experimental evidences

- P3L6

  ...over 2700 years... In the following section, the CCMI/CMIP6 models cover only around 200 years....how do you get 2700 years?

- P4L10

  "model output is compared to water vapor in the ERA5 reanalysis" - I agree that ERA5 has the best quality of the stratospheric water vapor if compared with other reanalyses...however, it is not an observed water vapor... Furthermore, water vapor observed in the stratosphere (e.g. MLS) is not used is the assimilation procedure of ERA5...Typically, all other reanalyses (ERA-Interim, JRA-55) have stratospheric water vapor that is not good enough for any scientific interpretation. Davis at al., ACP, 2019 writes: "...because of the known deficiencies in the representation of stratospheric transport in reanalyses, the stratospheric water vapor products from the current generation of reanalyses should generally not be used in scientific studies." The improvement of ERA5 (as documented in Wang et al., 2020) is probably a consequence of a better transport scheme...You should mention all these points. ERA5 H20 in the stratosphere is not the result of assimilated $H_2O$ observations but of transported $H_2O$ (+ contribution from methane oxidation). Especially there is not enough validation for the period 1979-1995, which was strongly influenced by volcanic eruption (El Chichon and Pinatubo)

- P4L17

  Few more details for the multi-linear regression of the QBO signal would be desirable. Typically two orthogonal components (zonal winds at 30 and 50 hPa) are used...

- P5L21-25

  It is not clear what is the advantage to show the results at 80 and 90 hPa...I would expect, to see the effect of slow upward propagation of the signal (like a taperecorder) so the signal at 80hPa should be slightly later than at 90 hPa (this can be seen in the satellite observations) ...but this slow propagation is typically not well reproduced by the models and reanalyses (which also use transport models to describe stratospheric H2O)...

- P6L19-20

  I miss here the citation: Bonazzola, M., and P. H. Haynes (2004), A trajectory-based study of the tropical tropopause region, J. Geophys. Res., 109, D20112, doi:10.1029/2003JD004356.

- P7L16

  MLR - multi-linear regression (?), this abbreviation was not explained, I think

- P7L24-29

  This is my strongest criticism: you consider ERA5 stratospheric H2O as "observation" (see L28). This is certainly not the case (see above). I would agree that temperature are more like "assimilated observations" but this is certainly not the case of stratospheric H2O in ERA5. I think, you should reformulate all these sentences and include a paragraph about stratospheric H2O in ERA5....Another point: are you using ERA5.0 that was shortly replaced by ERA5.1...it was recognized that temperatures (i.e. cold point tropopause) has a systematic bias for the period 2000-2009...this point should be also clarified.

- P9L5

  "ice lofting" - you are correctly mentioning that this process is not included in (all) models. This process is also not included into ERA5 stratospheric H2O what you consider as the "observation data"...at the end you compare different models with the ERA5 H2O also derived from the "internal" chemistry-transport model in the ERA5 (although the cold point temperatures may have a good quality).

---

## Referee Comment (RC2) · Anonymous Referee #2 · 22 Oct 2020

This study compares 12 models from the CCMI and CMIP6 projects with reanalysis observations to evaluate these models' skills in simulating ENSO's impacts on stratospheric water vapor variability at 90hPa in the tropics. One key metrics (asymmetry of the ENSO-water vapor relationship) is used in this work to assess each model's performance. It appears much effort has been put into this work and I don't see any serious problems in their analyses. In my view, the authors use right tools (resampling, composite) and their analyses are basically sound. However, considering that the authors seek to publish this study in a scientific journal, I expect to read more discussions to understand why some models are better than others and what possible causes of these failures and successes displayed in this paper could be. The authors only briefly dis-

cuss this issue in section 5 ( the final paragraph). They attribute the limitations of some "bad" models to weak interannual variability of water vapor around the troposphere and a lack of some key processes determining variability of cold point temperatures. I feel this very limited discussion is not sufficient and more in-depth thoughts are needed to improve the presentation and reasoning in the paper. So I consider that some minor revisions are required before accepting this article for publication.

To me, the main finding of the paper is that all models can well capture LN's impacts on water vapor variability in winter. I feel that the paper could benefit more from some more discussions on why all models perform better on this aspect, rather than just making a list of models with better performance.

In Fig4, there is a 2-month lag between T and water vapor and two models show very different patterns from others. I am wondering whether these differences are sensitive to the selection of the time lag. With different time lags, could we observe an improved performance in these two models.

The climatological mean state of vertical temperature profile in the tropics in models may play a key role to determine model's performance in replicating the ENSO-water vapor linkage. Here the authors mainly examine anomalies away from the mean state. I suggest that the authors should pay some attention on the mean state of cold point temperatures to examine whether some biases in the mean state could be translated to models' failures to reflect the ENSO-water vapor connection.

In my view, the selection of 15S to 15N in Fig. 3 needs to be justified. In addition, it is better to show the latitude -vertical transects of zonal mean temperature anomalies ( or an average across some longitudes in the pacific) to provide a 3-D picture of LN and EN' related tropical temperature responses I the 6 models.

---

## Short Comment (SC1) · 11 Nov 2020

Dear Chaim,

The manuscript presents an interest results but I think that some important points are still not clearly addressed in the manuscript. Please find below my comments and questions regarding these aspects. I am sure addressing these points will help improve the paper.

Kind regards,

Mohamadou

[Figure]

Major points:

1. Among the climate models used here some have interactive QBO (WACCM, HadGEM, . . .) and other have nudged QBO. This will lead to different modulations of the water vapor entry (tape recorder). Therefore, it is important to remove the QBO signal adequately in order to attribute properly the remaining variability to ENSO. However, this does not seem to be the case here or at least the description is not clear. Therefore, I have these questions:

a. Is the QBO proxy used in the MLR for all analyses calculated using the each model winds for the REF-C1 simulations?

b. Is the QBO proxy used in the MLR for all analyses coming from the observation (Berlin QBO or NASA)?

c. Is the QBO proxy used in the MLR for all analyses a combination of both a) for auto-generating QBO models and b) for nudged model?

If you have used the method a) or b) the results will seriously be questionable because of the time of the QBO modulation in the models, which auto-generate the QBO is different with the observed QBO signal,leading to bias results.

2. Regarding the Fig.1 where the non-linear response of water vapor induced by ENSO is claimed, it would be very interesting to see the QBO nudged models like (EMAC) regarding this non-linearity. Does ERA5 also show this non-linearity in Fig 1? The QBO contributes the most in the entry of water vapor anomalies via its modulation of the cold point temperature (Brinkop et al 2016; Diallo et al., 2018; Tao et al. 2019), therefore, it's important to handled it properly, which is seems to not be the case here.

3. It would be also very interesting show the tape recorder of each climate model simulation compared to ERA5. According to Hardiman et al. 2017, Figure 8, the HadGEM REF-C1 simulation compares very poorly with the SWOOSH observations, therefore, it would be interesting to see these models performance of tape recorder.

[Figure]

4. Thank you for adding the Tao et al, 2019 and Diallo et al 2018 into the discussion regarding the different contribution and modulation of the stratospheric water vapor and its entry by ENSO, QBO and seasonal cycle from observations (MLS, SWOOSH) and reanalyses.

References:

1) Hardiman, S. C., Butchart, N., O'Connor, F. M., and Rumbold, S. T.: The Met Office HadGEM3-ES chemistry–climate model: evaluation of stratospheric dynamics and its impact on ozone, Geosci. Model Dev., 10, 1209–1232, https://doi.org/10.5194/gmd-10-1209-2017, 2017.

2) Brinkop, S., Dameris, M., Jöckel, P., Garny, H., Lossow, S., andStiller, G.: The millennium water vapour drop in chemistry–climate model simulations, Atmos. Chem. Phys., 16, 8125–8140,https://doi.org/10.5194/acp-16-8125-2016, 2016.

3) Tao, M., Konopka, P., Ploeger, F., Yan, X., Wright, J. S., Diallo, M., Fueglistaler, S., and Riese, M.: Multitimescale variations in modeled stratospheric water vapor derived from three modern reanalysis products, Atmos. Chem. Phys., 19, 6509–6534, https://doi.org/10.5194/acp-19-6509-2019, 2019.

4) Diallo, M., Riese, M., Birner, T., Konopka, P., Müller, R., Hegglin, M. I., Santee, M. L., Baldwin, M., Legras, B., and Ploeger, F.: Response of stratospheric water vapor and ozone to the unusual timing of El Niño and the QBO disruption in 2015–2016, Atmos. Chem. Phys., 18, 13055–13073, https://doi.org/10.5194/acp-18-13055-2018, 2018.
* * *

---

## Author Comment (AC1) · 2 Dec 2020

**Reviewer #1**

This a very important and well-written paper and should be published by ACP. However, it contains some substantial inconsistencies related to the formulation of the results derived from pure model studies (like CCMI) or from models validated to some extent with observations (like the ERA5 reanalysis) or from pure observations (like MLS, HALOE or other satellite data). Thus, there are few major points which have to be clarified.

**We thank refer #1 for the positive review. Here we list the reply to the single points.**

Major points

• You use water vapor from the ERA5 reanalysis to validate the CCMI/CMIP6 models...you call it in the title of section 4: "Comparison to observation". In relation to the temperature fields, e.g. to the cold-point tropopause, ERA5 can be understood as an assimilated data set (like most other reanalyses), but this is certainly not the case in relation to the stratospheric water vapor. The stratospheric water vapor is a product derived from the ERA5 chemistry-transport module where, of course, the resolved cold-point tropopause plays a vary important important role. Thus, stratospheric water vapor in ERA5 is not assimilated with observations (like temperature), it is much more an almost "pure" model product.

Typically, all older reanalyses (ERA-Interim, JRA-55) have stratospheric water vapor that is not good enough for any scientific interpretation. Davis at al., ACP, 2019 writes: "...because of the known deficiencies in the representation of stratospheric transport in reanalyses, the stratospheric water vapor products from the current generation of reanalyses should generally not be used in scientific studies." The improvement of ERA5 (as documented in Wang et al., 2020) is probably a consequence of a better transport scheme...You should mention all these points. Even if the stratospheric water vapor in ERA5 is to some extent validated (Wang et al., 2020), there is still not enough validation of ERA5 for the period 1979-1995, which was strongly influenced by volcanic eruption (El Chichon and Pinatubo) and for which satellite observations are either not available or strongly disrupt by volcanic aerosol. Finally, it is not clear if ERA5.0 or ERA5.1 is used (it is also not clear in Wang et al., 2020). The latter version removes a significant temperature bias of the cold point tropopause for the period 2000-2009 (see Simmons et al., 2020, ECMWF, Technical Memo 859).

**We agree that there is not much validation and few observations to constrain ERA-5 before 1995, and hence before 1995 ERA-5 water vapor should be treated with caution. Given this, the added value in using ERA5 is minimal as compared to e.g. SWOOSH (Stratospheric Water and OzOne Satellite Homogenized), and hence we now use SWOOSH in the paper. Results are essentially unchanged.  We now use ERA5.1 for cold point temperature only.**

• Your study shows that there are models which do not lead to a moistening of the stratosphere after El Nino events (like last 3 models shown in Fig. 5). However, moistening of the stratosphere after strong EL Nino is the best documented and validated finding (Geller et al., 2002, Scaife et al.,

2003, Randel et al., 2009, Konopka et al., 2016) although as pointed by Garfinkel et al. 2013, there are some differences in the intensity of such moistening which depend on maximum relative to the winter season and relative to El Nino's location (Central or Eastern Pacific). I think that this point (models do not represent moistening after EL Nino correctly) should be mentioned in the abstract. I think that this point is at least equally important as your statement related to the non-linear behavior between La Nina and El Nino or that in all models La Nina leads to moistening in winter relative to neutral ENSO. Both statements, even very interesting, are derived "only" from the models and only partially present in ERA5 reanalysis.

**We have created a version of figure 7 but for a threshold of 1.5C as opposed to 1C for both CMIP and CCMI. See below. Most models fail to reproduce the enhanced water vapor in the spring after El Nino. The only exceptions are WACCM, CAM, GISS, GFDL, and the Met Office model in CMIP6. We have added mention of this to the abstract.**

entry water vapor and ENSO

entry water vapor and ENSO

H2Oanom entry, 15S–15N:

(a) SWOOSH, N=27

(b) WACCM

(c) CAM4Chem

(d) HadGEM3-ES

(e) MRI-ESM1r1

(f) NIWA-UKCA

(g) EMAC-L47MA

EN  LN
Neutral

Minor comments:

• P2L3 ...is around half of that for global mean surface albedo OR cloud feedbacks... (I think, this is what the the cited papers show)

**Yes, corrected to "or"**

 • P2L28 I think, it is not correct to use the Brinkop et al., 2016 citation to support the idea of nonlinear effects....what they showed is that the millennium drop was a combination of El Nino (i.e. wet phase in the stratosphere) followed by La Nina (i.e. dry phase in the stratosphere) with QBO being in the east phase (i.e. enhancing the dry phase)...This is a very linear interpretation without any nonlinear effects

**We removed this.**

• P2L30 For the non-linear effect discussed in Garfinkel et al., 2018 you should also mention that this is a pure model study without any experimental evidences

**Actually, Garfinkel et al did compare the model results (which indeed were the main focus of that paper) with observational data. See figure 4 and 9 in that study. The same effect was seen in observational data, but the limited record length limits the certainty of the conclusions reached in that study using the observational data. This has now been clarified in the present paper.**

**In any event, we now show the nonlinear effect in SWOOSH again in the revised Figure 1 of this paper (repeated below).**

[Figure]

**Figure 1.** Anomalous 80hPa water vapor in WACCM as compared to the value of the Nino3.4 index for (a) November and December; (b) January and February; (c) March and April; (d) May and June. Each dot corresponds to one model-year. When a polynomial fit better describes the dependence on ENSO than a linear fit, we show the $R^2$ for a linear fit and adjusted $R^2$ for the polynomial fit (see section 2.2). Otherwise we show a linear least-squares best fit in each panel.

• P3L6 ...over 2700 years... In the following section, the CCMI/CMIP6 models cover only around 200 years....how do you get 2700 years?

**There are six CMIP6 integrations, each of 164 years. The CCMI integrations are between 138 and 140 years each (depending on the precise end date), and for NIWA and the NCAR models there are multiple ensemble members. The total sum is of over 2700 numerically computed years stored as model output.**

**We have added more details to Table 1 to clarify this.**

• P4L10 "model output is compared to water vapor in the ERA5 reanalysis" - I agree that ERA5 has the best quality of the stratospheric water vapor if compared with other reanalyses...however, it is not an observed water vapor... Furthermore, water vapor observed in the stratosphere (e.g. MLS) is not used is the assimilation procedure of ERA5...Typically, all other reanalyses (ERA-Interim, JRA-55) have stratospheric water vapor that is not good enough for any scientific interpretation. Davis at al., ACP, 2019 writes: "...because of the known deficiencies in the representation of stratospheric transport in reanalyses, the stratospheric water vapor products from the current generation of reanalyses should generally not be used in scientific studies." The improvement of ERA5 (as documented in Wang et al., 2020) is probably a consequence of a better transport scheme...You should mention all these points. ERA5 H20 in the stratosphere is not the result of assimilated $H_2O$ observations but of transported $H_2O$ (+ contribution from methane oxidation). Especially there is not enough validation for the period 1979-1995, which was strongly influenced by volcanic eruption (El Chichon and Pinatubo)

**As discussed earlier, we now use measurements of water vapor from the SWOOSH data set.**

• P4L17 Few more details for the multi-linear regression of the QBO signal would be desirable. Typically two orthogonal components (zonal winds at 30 and 50 hPa) are used...

**We agree that if water vapor higher up is considered, a second QBO index is needed, or a different lag needs to be chosen. Nevertheless, for water vapor at the cold point adding a second QBO has little effect, as QBO at 50hPa with a 1-2 month lag maximizes the effect. This effect is shown explicitly in Tan et al 2019, now cited. This point has now been clarified.**

• P5L21-25 It is not clear what is the advantage to show the results at 80 and 90 hPa...I would expect, to see the effect of slow upward propagation of the signal (like a tape recorder) so the signal at 80hPa should be slightly later than at 90 hPa (this can be seen in the satellite observations) ...but this slow propagation is typically not well reproduced by the models and reanalyses (which also use transport models to describe stratospheric H2O)...

**We have replaced the 90hPa column with the SWOOSH observational product.**

• P6L19-20 I miss here the citation: Bonazzola, M., and P. H. Haynes (2004), A trajectory based study of the tropical tropopause region, J. Geophys. Res., 109, D20112, doi:10.1029/2003JD004356.

**added**

• P7L16 MLR - multi-linear regression (?), this abbreviation was not explained, I think

**Abbreviation now added earlier**

• P7L24-29 This is my strongest criticism: you consider ERA5 stratospheric H2O as "observation" (see L28). This is certainly not the case (see above). I would agree that temperature are more like "assimilated observations" but this is certainly not the case of stratospheric H2O in ERA5. I think, you should reformulate all these sentences and include a paragraph about stratospheric H2O in ERA5....Another point: are you using ERA5.0 that was shortly replaced by ERA5.1...it was recognized that temperatures (i.e. cold point tropopause) has a systematic bias for the period 2000-2009...this point should be also clarified.

**See above with regards to water vapor. We use ERA5.1 for cold point temperatures, now clarified.**

• P9L5 "ice lofting" - you are correctly mentioning that this process is not included in (all) models. This process is also not included into ERA5 stratospheric H2O what you consider as the "observation data"...at the end you compare different models with the ERA5 H2O also derived from the "internal" chemistry-transport model in the ERA5 (although the cold point temperatures may have a good quality).

**See above**

**Reviewer #2**

This study compares 12 models from the CCMI and CMIP6 projects with reanalysis observations to evaluate these models' skills in simulating ENSO's impacts on stratospheric water vapor variability at 90hPa in the tropics. One key metrics (asymmetry of the ENSO-water vapor relationship) is used in this work to assess each model's performance. It appears much effort has been put into this work and I don't see any serious problems in their analyses. In my view, the authors use right tools (resampling, composite) and their analyses are basically sound. However, considering that the authors seek to publish this study in a scientific journal, I expect to read more discussions to understand why some models are better than others and what possible causes of these failures and successes displayed in this paper could be. The authors only briefly dis cuss this issue in section 5 ( the final paragraph). They attribute the limitations of some "bad" models to weak interannual variability of water vapor around the troposphere and a lack of some key processes determining variability of cold point temperatures. I feel this very limited discussion is not sufficient and more in-depth thoughts are needed to improve the presentation and reasoning in the paper. So I consider that some minor revisions are required before accepting this article for publication.

**We thank reviewer #2 for the positive and constructive comments.**

To me, the main finding of the paper is that all models can well capture LN's impacts on water vapor variability in winter. I feel that the paper could benefit more from some more discussions on why all models perform better on this aspect, rather than just making a list of models with better performance.

**To improve the discussion and the results' interpretation, we have added a figure to the revised manuscript comparing the cold point tropopause temperature to ERA5.1 for all models. While the cold point temperature can account for the bias in some models, it doesn't for others. Specifically, MRI simulates more water vapor variability in CMIP6 than CCMI even as the cold point has cooled. The new figure is copied below**

[Figure]

[Figure]

**Figure 8.** Climatological mean temperature in January and February from 10S-10N for each model and model level data from ERA5.1. Panel b enlarges the cold point region on panel a. For CMIP6 models, we only show the 100hPa and 70hPa due to the limited resolution available in the CMIP data archive.

In Fig4, there is a 2-month lag between T and water vapor and two models show very different patterns from others. I am wondering whether these differences are sensitive to the selection of the time lag. With different time lags, could we observe an improved performance in these two models.

**We have recreated this figure using a 1 and 3 month lag. Results are very similar (which is not surprising at least to us considering the high autocorrelation of sea surface temperature anomalies). There truly is a wide discrepancy among the models.**

The climatological mean state of vertical temperature profile in the tropics in models may play a key role to determine model's performance in replicating the ENSO-water vapor linkage. Here the authors mainly examine anomalies away from the mean state. I suggest that the authors should pay some attention on the mean state of cold point temperatures to examine whether some biases in the mean state could be translated to models' failures to reflect the ENSO-water vapor connection.

**We thank the referee for the suggestion. We have added a figure showing the climatological cold point in these models (see above). Most models have a warm bias (evident in CMIP5 too), and for some models the bias in variability can be explained by the bias in the cold point climatology. This figure will be included in the revised manuscript.**

In my view, the selection of 15S to 15N in Fig. 3 needs to be justified. In addition, it is better to show the latitude -vertical transects of zonal mean temperature anomalies ( or an average across some longitudes in the pacific) to provide a 3-D picture of LN and EN' related tropical temperature responses I the 6 models.

**As discussed as far back as Gettleman et al 2001 and also in detail in Garfinkel et al 2013 (both cited in the paper), ENSO leads to opposite signed temperature anomalies in the Central and West Pacific, and the cold point moves east for El Nino and west for La Nina. Hence a zonal mean picture, or averaging over a fixed longitude range, isn't enlightening. Table 2 in the present study makes this point as well.**

**We have created Figure 3 for 10N-10S and also 20N-20S, and results are essentially identical.**

**In the revised submission we will include lat vs. lon maps of temperature anomalies at 90hPa (for CCMI) and 100hPa (for CMIP6) for these models in the supplemental material.**

**Reviewer #3, Mohamadou Diallo**

Major points: 1. Among the climate models used here some have interactive QBO (WACCM, HadGEM, . . .) and other have nudged QBO. This will lead to different modulations of the water vapor entry (tape recorder). Therefore, it is important to remove the QBO signal adequately in order to attribute properly the remaining variability to ENSO. However, this does not seem to be the case here or at least the description is not clear. Therefore, I have these questions: a. Is the QBO proxy used in the MLR for all analyses calculated using the each model winds for the REF-C1 simulations? b. Is the QBO proxy used in the MLR for all analyses coming from the observation (Berlin QBO or NASA)? c. Is the QBO proxy used in the MLR for all analyses a combination of both a) for auto-generating QBO models and b) for nudged model? If you have used the method a) or b) the results will seriously be questionable because of the time of the QBO modulation in the models, which auto-generate the QBO is different with the observed QBO signal,leading to bias results.

**We use the QBO at 50hPa from each data source individually. This has been clarified in the revised version of the manuscript.**

2. Regarding the Fig.1 where the non-linear response of water vapor induced by ENSO is claimed, it would be very interesting to see the QBO nudged models like (EMAC) regarding this non-linearity. Does ERA5 also show this non-linearity in Fig 1? The QBO contributes the most in the entry of water vapor anomalies via its modulation of the cold point temperature (Brinkop et al 2016; Diallo et al., 2018; Tao et al. 2019), therefore, it's important to handled it properly, which is seems to not be the case here.

**Figure 4 of Garfinkel et al 2018 shows the nonlinearity in SWOOSH water vapor, and a similar effect is evident in ERA-5 though the limited observational record means that the effect is not robust and could be due to sampling variability. Following the comments of the reviewer #1, we now include SWOOSH instead of ERA-5 in the present manuscript, and include SWOOSH in Figure**

**1. Despite the observational nonlinearity of the water vapor response to the ENSO signal, most models don't show this nonlinearity, as discussed in the text, though the NCAR models do.**

3. It would be also very interesting show the tape recorder of each climate model simulation compared to ERA5. According to Hardiman et al. 2017, Figure 8, the HadGEM REF-C1 simulation compares very poorly with the SWOOSH observations, therefore, it would be interesting to see these models performance of tape recorder.

**While this would indeed be interesting, it is beyond the scope of this paper to consider water vapor variability in these models above the cold point. We have added to the discussion section that future work is needed to understand the impact of these changes in the lowermost stratosphere on changes in water vapor higher up.**

4. Thank you for adding the Tao et al, 2019 and Diallo et al 2018 into the discussion regarding the different contribution and modulation of the stratospheric water vapor and its entry by ENSO, QBO and seasonal cycle from observations (MLS, SWOOSH) and reanalyses.

**added**

References: 1) Hardiman, S. C., Butchart, N., O'Connor, F. M., and Rumbold, S. T.: The Met Office HadGEM3-ES chemistry–climate model: evaluation of stratospheric dynamics and its impact on ozone, Geosci. Model Dev., 10, 1209–1232, https://doi.org/10.5194/gmd10-1209-2017, 2017. 2) Brinkop, S., Dameris, M., Jöckel, P., Garny, H., Lossow, S., andStiller, G.: The millennium water vapour drop in chemistry–climate model simulations, Atmos. Chem. Phys., 16, 8125–8140,https://doi.org/10.5194/acp-16-8125-2016, 2016. 3) Tao, M., Konopka, P., Ploeger, F., Yan, X., Wright, J. S., Diallo, M., Fueglistaler, S., and Riese, M.: Multitimescale variations in modeled stratospheric water vapor derived from three modern reanalysis products, Atmos. Chem. Phys., 19, 6509–6534, https://doi.org/10.5194/acp-19-6509-2019, 2019. 4) Diallo, M., Riese, M., Birner, T., Konopka, P., Müller, R., Hegglin, M. I., Santee, M. L., Baldwin, M., Legras, B., and Ploeger, F.: Response of stratospheric water vapor and ozone to the unusual timing of El Niño and the QBO disruption in 2015–2016, Atmos. Chem. Phys., 18, 13055–13073, https://doi.org/10.5194/acp-18-13055-2018, 2018.

---

## Referee Report (RR1)

Second review of the paper:

"Influence of ENSO on entry stratospheric water vapor..."

written by Chaim I Garfinkel et al.

**General:**
The paper improved. However there is still one important point which needs clarification. See below.

**Major points**

- You write in your abstract: "the only aspect of the entry water vapor response with consensus is that La Nina leads to moistening in winter relative to neutral ENSO". What you see is only a positive correlation between the moistening at 82 hPa and the La Nina index at the surface! You should take into account that the propagation of the La Nina signal (e.g. from the main convective outflow around 200 hPa) to the lower stratosphere around 80 hPa needs time of the order of few months (tape-recorder). Typically, this time is between 3 and 5 months so your interpretation that the enhanced water vapor in the lower stratosphere at the begin of the boreal winter is caused by La Nina is not correct. This error repeats many times in your paper, especially in relation to your non-linear correlation between the ENSO index and water vapor in the lower stratosphere. The correlation itself does not explain the formation of this relation!

---

## Author Response (AR2)

Second review of the paper: "Influence of ENSO on entry stratospheric water vapor…" written by Chaim I Garfinkel et al.

General: The paper improved. However there is still one important point which needs clarification. See below.

Major points

You write in your abstract: "the only aspect of the entry water vapor response with consensus is that La Nina leads to moistening in winter relative to neutral ENSO". What you see is only a positive correlation between the moistening at 82 hPa and the La Nina index at the surface! You should take into account that the propagation of the La Nina signal (e.g. from the main convective outflow around 200 hPa) to the lower stratosphere around 80 hPa needs time of the order of few months (tape-recorder). Typically, this time is between 3 and 5 months so your interpretation that the enhanced water vapor in the lower stratosphere at the begin of the boreal winter is caused by La Nina is not correct. This error repeats many times in your paper, especially in relation to your non-linear correlation between the ENSO index and water vapor in the lower stratosphere. The correlation itself does not explain the formation of this relation!

We certainly are aware that transport to 80hPa is not immediate, and we apologize if we were not sufficiently clear and implied otherwise. Note however that the characteristic timescale of ENSO is much slower than this transport timescale. ENSO events typically develop in the preceding early fall and peak in early winter, such that La Nina conditions are present well before the January/February signal we are ascribing to La Nina. Hence the statement in the abstract, and similar statements elsewhere in the text, are actually correct.

However we realize this was confusing, and we have therefore added to the methods section:

"A typical ENSO event slowly strengthens in the summer and fall, reaches its maximum strength in late fall or early winter, and then decays in the spring (Figure 1 of Wang and Fiedler, 2006). This evolution is captured in the models (Supplemental Figure 1). While the influence of ENSO on tropospheric temperatures is rapid due to convection, there is a few month lag in transport from the level with peak convective outflow to the cold point (Mote et al., 1996; Fueglistaler et al., 2004). However the sea surface temperature anomalies due to ENSO are already established by fall, and hence all of the anomalies shown here can be associated with ENSO"

This new figure is included in the supplement.

[Figure]

The reviewer asks about a correlation between an ENSO index and entry water vapor, but we aren't sure what figure the reviewer is referring to as we don't show any correlations between the ENSO index and water vapor in the lower stratosphere anywhere in this paper. Our guess is that the reviewer was referring to figure 4, which showed the correlation between near surface temperature and temperature at 500hPa with entry water near 80hPa. (This is the only figure based on correlations anywhere in our paper.) We used a two month lag for this figure, and it is reasonable for the reviewer to ask whether a 4 month lag is more appropriate. To that end, we show below the same figure but using a 4 month lag, that is, January and February temperature with May and June entry water. Results are the same for five of six models, and this one exception has no impact on our overall points: instead of 4 models showing one signal vs. 2 showing a different one, now 3 models show one signal and 3 show a different one. This new figure is included in the supplement